# Combining an antibiotic stewardship program with a 15-pathogen viral panel to reduce inappropriate antibiotic prescribing

Cosby G. Arnold,[1] Tyra Furtado,[2] Heejung Bang,[3] Glenn Harnett,[4] Larissa S. May[1]

**ABSTRACT** Acute upper respiratory tract infections (ARI) frequently result in unnecessary antibiotic prescribing in outpatient settings, contributing to antibiotic resistance and avoidable adverse outcomes. While antimicrobial stewardship programs (ASPs) have shown promise, their implementation in urgent care settings remains limited. The impact of point-of-care (POC) respiratory viral panels on antibiotic prescribing is also uncertain. This study evaluates the effect of the BIOFIRE SPOTFIRE Respiratory Panel, a 15-pathogen panel, with integrated ASP interventions on antibiotic prescribing in a high-volume urgent care setting in the Southeast United States. This was a prospective cohort study of adults with ARI who presented to an urgent care in Louisiana from June to August 2024. A randomly selected, seasonally matched, historic usual care group served as a control group. We compared the impact of BIOFIRE SPOTFIRE Respiratory Panel testing and ASP on antibiotic prescribing. A total of 296 patients were prospectively enrolled, with 600 randomly selected historical controls. The intervention group had a significantly lower antibiotic prescribing rate (24.3% vs 38.2%; adjusted odds ratio (aOR) 0.50, 95% confidence interval (CI) 0.36–0.68) and inappropriate antibiotic use rate (15.9% vs 30.8%; aOR 0.39, 95% CI 0.28–0.57) compared to controls. Steroid prescribing also decreased (17.9% vs 29.0%; aOR 0.55, 95% CI 0.39–0.78). Implementation of a POC multirespiratory pathogen molecular test combined with an ASP intervention significantly reduced antibiotic and steroid prescribing in a high-volume urgent care setting.

**IMPORTANCE** Acute upper respiratory tract infections are common and often result in unnecessary antibiotic prescribing. We found a significant reduction in inappropriate antibiotic use after implementing an antibiotic stewardship program with a point-of-care respiratory viral panel in an urgent care setting.

**KEYWORDS** antibiotic stewardship, upper respiratory tract infections, point-of-care testing

A cute upper respiratory tract infections (ARIs) are commonly seen in ambulatory healthcare settings and often result in unnecessary antibiotic prescribing (1). Approximately 60% of all antimicrobial use occurs in the outpatient setting, with 28% of antibiotics inappropriately prescribed (1, 2). Antimicrobial overuse contributes to the emergence and spread of drug-resistant bacteria, potentially avoidable adverse drug reactions, and poor clinical outcomes (3, 4). Although antimicrobial stewardship programs (ASPs) have been shown to reduce inappropriate prescribing (5), ASPs targeted to the outpatient setting are uncommon (6).

Studies on the impact of rapid respiratory viral panels on antibiotic use in the emergency department (ED) and ambulatory healthcare settings have produced mixed results, with some suggested benefit of influenza and SARS-CoV-2 testing (7–10).

Address correspondence to Cosby G. Arnold, cgarnold@ucdavis.edu.

L.S.M. has received honoraria from bioMérieux for participation on advisory boards and for speaking. G.H. has received honoraria from bioMérieux for speaking.

See the funding table on p. 7.

Whether viral testing, performed at the point-of-care (POC), decreases prescribing when integrated with an ASP remains unknown. Therefore, we sought to prospectively evaluate the effects of POC multirespiratory pathogen molecular testing with antibiotic stewardship implementation compared to usual care (POC influenza and SARS-CoV-2 antigen testing) on antibiotic prescribing.

## MATERIALS AND METHODS

### Study design and participants

We conducted a prospective cohort study of adults with ARI who presented to a non-academic, community urgent care center from June 2024 to August 2024. The urgent care is in Baton Rouge, Louisiana, and has 20,000 encounters per year. The study was approved by a central Institutional Review Board (WCG IRB).

Patients were eligible for inclusion if they were 18 years of age or older and the urgent care provider planned to order a standard POC viral antigen panel for COVID-19 and/or influenza (consistent with a real-world setting) testing. Subjects were screened if they presented to the clinic with respiratory symptoms, and the clinician or staff (based on clinic standing orders) planned to order a standard POC viral antigen test. If the patient chose to enroll in the study, the clinic staff ordered the BIOFIRE SPOTFIRE Respiratory Panel instead of the standard test. All testing was performed by the clinic's staff. The results were scanned into the electronic health records (EHR) and attached to the patient's visit in the EHR. No standard viral testing was performed on subjects enrolled. If the clinician planned to order the viral antigen testing on a patient, that patient was screened and offered enrollment regardless of the chief complaint.

We excluded return visits within 72 hours and patients who were unable to provide informed consent in English. A historic usual care group served as a control group. This group was randomly selected from all patients seen for ARI, matched by month to control for seasonality, in the prior year.

### Study procedures

After obtaining informed consent, trained research assistants collected a nasopharyngeal swab sample for BIOFIRE SPOTFIRE Respiratory Panel testing. Routine clinical data were obtained from the EHR. Trained and supervised data extractors performed a focused manual chart review to obtain key variables and outcome information not obtained through electronic capture. In addition, charts were abstracted for outcomes where available.

### Implementation science approach

#### Pre-implementation phase (1 month, April 2024)

We sought to establish organizational support for this project. First, we created a local implementation team comprising a dedicated clinical champion, research coordinator, and informatics support. Next, we conducted a context assessment to identify current prescribing practices. We used provider surveys, structured query language data query, and manual chart review to examine baseline prescribing practices among urgent care providers. As part of the context assessment, we examined stewardship intervention preferences and workflow fit. We used the workflow analysis and context assessment to inform the intervention selection from the MITIGATE toolkit (11).

We approached all eligible providers, nurses, and administrators via email to participate in an anonymous provider survey. Individual providers, nurses, and administrators were invited to participate in in-depth semi-structured interviews.

The provider survey and stakeholder interview guides were based on prior survey work in this area, including our group's project with the Centers for Disease Control (CDC) (12, 13). All qualitative data collection instruments were finalized through a collaboration

with study team experts and local site stakeholders during the pre-implementation phase.

Seven providers had pre-implementation interviews, and all seven providers completed pre- and post-implementation surveys. Four medical assistants and one clinic manager had pre-implementation interviews. CDC antibiotic stewardship posters were posted around the urgent care clinic. Each provider also signed a CDC antibiotic stewardship commitment letter. All providers and clinic staff received CDC Get Smart Buttons and attended an online antibiotic stewardship webinar. The site PI also conducted stakeholder interviews with the Chief Medical Officer, all providers, and four staff members. All components of the intervention were based on the Mitigate toolkit.

### Adaptation phase (1 month, May 2024)

This phase focused on adapting and finalizing the implementation plan for roll-out to the providers. All providers received a provider-specific multifaceted stewardship intervention consistent with CDC core elements for outpatient antimicrobial stewardship: commitment, action, monitoring, reporting, and education, as well as a behavioral component that uses public commitment. Interventions from the MITIGATE toolkit were locally adapted and included site commitment and patient education (e.g., discharge instructions and waiting room educational materials). For departmental public commitment, we used the CDC's "Commitment Letter to Our Patients" template. These were placed in waiting rooms along with laminated patient education materials. Providers also received information on the viruses detected by the SPOTFIRE test and tools for patient communication.

## Outcomes

Our primary outcomes were the number of participants receiving prescriptions for antibiotics and steroids after intervention as well as the number receiving an inappropriate antibiotic. Inappropriate prescribing was defined as an antibiotic prescription without evidence of bacterial infection based on a review of final ICD-10 diagnosis, laboratory testing, and chest radiograph (if performed) results. As secondary outcomes, we measured the number of respiratory pathogens identified by the SPOTFIRE test, chest radiograph utilization, symptom resolution, and unscheduled medical visits within 7 days.

## Statistical analysis

We estimated that a sample size of 300 participants post-intervention and 600 participants pre-intervention would provide >80% power to detect a 10 percent decrease (in absolute scale) in antibiotic use between the two groups, based on a baseline prescribing rate of 50%.

We used the $\chi^2$ or Fisher's exact statistic for analysis of categorical variables and Student's $t$-tests for continuous variables. We used multiple logistic regression models for binary outcomes, where adjusted odds ratio (aOR) was estimated with the 95% confidence interval (CI). Model covariates were selected based on prior literature and investigator opinion of clinical importance and relevance; we did not perform data-driven variable selection. Additionally, we estimated the (unadjusted) risk difference with 95% CI.

We used descriptive statistics to summarize data, such as mean and standard deviation (SD) for continuous variables and frequency (%) for categorical variables, and Likert scales for pre- and post-intervention survey results. These data were quantified with frequency counts to identify primary themes. Study data were managed in RedCap. All analyses were performed in SAS version 9.4 (SAS Institute, Cary, NC).

## RESULTS

A total of 300 patients were prospectively enrolled in our study. Four withdrew consent and were excluded, leaving 296 for analysis. Mean age (± SD) was 44.5 ± 17 years, and 223 (75.3%) were female. Most participants were Black (72.6%) and non-Hispanic (92.2%). Among the 600 controls, mean age was 41.4 ± 15 year, and 444 (74.0%) were female. The most common race was Black (66.3%), and 94.3% were non-Hispanic. Table 1 describes patient characteristics.

Patients in the post-intervention cohort were less likely to receive antibiotics (aOR 0.50; 95% CI 0.36–0.68; $P < 0.0001$). The rate of antibiotic prescribing was 38.2% prior to intervention and 24.3% post-intervention; a difference of 13.9% (95% CI 7.4%–20.3%). Inappropriate prescribing also decreased after intervention (aOR 0.40; 95% CI 0.28–0.57; $P < 0.0001$). Patients were also less likely to receive steroids post-intervention (aOR 0.55; 95% CI 0.39–0.78; $P = 0.0008$). The rate of steroid prescribing was 29.0% pre-intervention vs 17.9% post-intervention; difference of 11.1% (95% CI 5.2%–17.0%); see Table 2. The total number of patients with a positive SPOTFIRE test result was 154 (52%), and five (3.2%) of these patients tested positive for more than one pathogen. SPOTFIRE test results are provided in Table 3. Prior to intervention, 12 (2%) chest radiographs were ordered, and 2 (16.7%) were positive for pneumonia. Post-intervention, 15 (5.1%) chest radiographs were ordered, and seven (46.7%) of these were positive. Although the trends observed in post-intervention ordering behavior appear encouraging, we did not evaluate for statistical significance given the small sample size.

In the 7-day follow-up interview, 65 (32%) participants reported taking antibiotics and, among these, 40 (62%) reported completing their antibiotic prescription. Most

**TABLE 1** Characteristics of the study cohort[b,c,d]

| | Before intervention (N = 600) | After intervention (N = 296) | P-value |
|---|---|---|---|
| Age (in y), mean (SD), range | 41.4 (15.0), 18–88 | 44.5 (17.0), 18–91 | 0.01 |
| Sex | | | |
| Female | 444 (74.0%) | 223 (75.3%) | 0.67 |
| Male | 156 (26.0%) | 73 (24.7%) | |
| Ethnicity | | | |
| Hispanic/Latino | 17 (2.8%) | 12 (4.1%) | 0.47 |
| Not Hispanic/Latino | 566 (94.3%) | 273 (92.2%) | |
| Declined to answer | 17 (2.8%) | 11 (3.7%) | |
| Race | | | |
| Asian | 8 (1.3%) | 4 (1.4%) | 0.17[a] |
| Black or African American | 398 (66.3%) | 215 (72.6%) | |
| White | 170 (28.3%) | 63 (21.2%) | |
| American Indian/Alaskan Native | 1 (0.2%) | 0 (0%) | |
| Declined to answer | 23 (3.8%) | 14 (4.7%) | |
| Comorbidities | | | |
| Diabetes mellitus | 72 (12.0%) | 51 (17.2%) | 0.03 |
| Hypertension | 176 (29.3%) | 108 (36.5%) | 0.03 |
| Coronary artery disease | 1 (0.2%) | 1 (0.3%) | 0.55[a] |
| Congestive heart failure | 2 (0.3%) | 0 (0%) | 1.00[a] |
| Asthma | 26 (4.3%) | 17 (5.7%) | 0.35 |
| COPD | 1 (0.2%) | 2 (0.7%) | 0.26[a] |
| Other respiratory conditions | 34 (5.7) | 24 (8.1%) | 0.16 |
| Medication | | | |
| Immunosuppressant use | 21 (3.5%) | 3 (1.0%) | 0.03 |

[a]Fisher exact test due to low expected cell count. For other categorical variables, the $\chi^2$ test was used.
[b]COPD, chronic obstructive pulmonary disease; SD, standard deviation.
[c]Categorical variables are shown as frequency (%).
[d]For age, t-test with unequal variance was used.

**TABLE 2** Primary outcome results, Adjusted for Clinical Factors (N = 896)[a,b]

| Clinical factor | aOR (95% CI) | *P*-value |
|---|---|---|
| Prescribing antibiotics | | |
| Post-intervention | 0.50 (0.36, 0.68) | <0.0001 |
| Age (per 1 year) | 1.01 (1.00, 1.02) | 0.043 |
| Diabetes | 1.15 (0.76, 1.75) | 0.504 |
| Respiratory condition (asthma, COPD, or other) | 1.46 (0.84, 2.55) | 0.178 |
| Immunosuppressants | 1.39 (0.61, 3.17) | 0.440 |
| Prescribing steroids | | |
| Post-intervention | 0.55 (0.39, 0.78) | 0.0008 |
| Age (per 1 year) | 1.00 (0.99, 1.01) | 0.978 |
| Diabetes | 0.63 (0.38, 1.05) | 0.073 |
| Respiratory condition (asthma, COPD, or other) | 1.34 (0.74, 2.44) | 0.335 |
| Immunosuppressants | 2.23 (0.97, 5.10) | 0.058 |
| Prescribing antibiotics inappropriately | | |
| Post-intervention | 0.40 (0.28, 0.57) | <0.0001 |
| Age (per 1 year) | 1.01 (1.00, 1.02) | 0.075 |
| Diabetes | 1.22 (0.78, 1.91) | 0.376 |
| Respiratory condition (asthma, COPD, or other) | 1.92 (1.09, 3.40) | 0.025 |
| Immunosuppressants | 1.37 (0.58, 3.23) | 0.470 |

[a]CI, confidence interval; aOR, adjusted odds ratio.
[b]Covariates adjusted are prespecified based on the scientific rationale. We did not perform data-driven variable selection.

patients (79%) reported resolution of symptoms, and 9% reported unplanned medical visits. See Table 4 for interview results.

Among 147 clinicians participating in a post-intervention survey, 4 (3%) selected one or 2, 12 (8%) selected 3, and 131 (89%) selected four or 5 on a 5-point Likert scale rating their confidence (1, less confident; 5, more confident) in not prescribing antibiotics after SPOTFIRE results.

## DISCUSSION

In this study, we used an implementation science approach to evaluate the effect of POC multiplex testing combined with an antibiotic stewardship program on antibiotic prescribing practices. We observed a significant reduction in antibiotic prescriptions after this intervention.

The MITIGATE toolkit provides a step-by-step evidence-based implementation guide for ED and urgent care settings and has been successfully adapted to ambulatory care (14, 15) and urgent care clinics (5). However, to our knowledge, the current study is the first to demonstrate the utility of implementing MITIGATE in the context of multiplex POC testing for respiratory pathogen testing.

**TABLE 3** BIOFIRE SPOTFIRE respiratory panel positive results percent distribution by pathogen[b]

| Pathogen | N (%)[a] |
|---|---|
| Adenovirus | 1 (0.65%) |
| Seasonal coronavirus | 8 (5.20%) |
| COVID-19 | 101 (65.58%) |
| Parainfluenza | 3 (1.95%) |
| Epstein-Barr virus | 1 (0.65%) |
| Metapneumovirus | 3 (1.95%) |
| Influenza A | 1 (0.65%) |
| Mycoplasma pneumonia | 1 (0.65%) |
| Rhinovirus/enterovirus | 40 (25.97%) |

[a]Frequency (N) and percent (%).
[b]Percentages calculated based on 154 total positive tests (subjects could have >1 positive test on respiratory panel).

**TABLE 4** Results of 7-day patient follow-up interview (N = 206)[b]

| Question | N (%)[a] |
|---|---|
| Took antibiotics | 65 (32%) |
| Completed antibiotic prescription (if took antibiotics) | 40 (62%) |
| Took antivirals | 7 (3%) |
| Completed antiviral prescription (if took antivirals) | 5 (71%) |
| Symptoms resolved | 163 (79%) |
| Reported complications or adverse response to medication | 1 (0.5%) |
| Reported unplanned medical visit(s) | 18 (9%) |

[a]Frequency (N) and percent (%).
[b]A total of 206 of 296 participants provided responses.

Implementation of antibiotic stewardship interventions for ARIs in ED and urgent care settings is feasible and effective. In prior work, we found that intensive behavioral nudging methods did not impact prescribing behavior in academic settings, possibly because antibiotic-inappropriate prescribing was already uncommon given the lower prescribing rates in these settings, perhaps due to the presence of trainees and evidence-based practices (12, 13). However, implementation of these programs in a large rural healthcare system with high antibiotic prescribing rates resulted in a significant reduction in inappropriate prescribing for ARIs (5).

Our findings align with those of prior work demonstrating the potential role of POC testing in mitigating inappropriate antibiotic use, ultimately contributing to improved antimicrobial stewardship. Cohen et al. reported a 36% reduction in antibiotic prescriptions for pharyngitis after the implementation of rapid streptococcal antigen testing in urgent care settings (16). Zhang et al. reported a 24% decrease in antibiotic use when C-reactive protein testing was employed to guide treatment decisions for lower respiratory tract infections (17). In a meta-analysis, multiplex testing reduced time to results, improved antiviral use, and reduced ED and hospital length of stay (18).

Despite its advantages, integrating POC testing into routine clinical practice presents several challenges. Antimicrobial prescribing is a complex behavior that is influenced by a combination of factors. Aside from individual provider characteristics (e.g., knowledge, skills, memory, attention, and decision-making), there are external influences (e.g., peer comparison) that differ by context and setting (19). Financial constraints, including the cost of POC devices and reimbursement issues, remain significant barriers. Additionally, clinician adherence and confidence in POC-guided decision-making can vary, with some providers expressing concerns about test accuracy and reliability (20), although PCR-based POC testing is highly accurate (21). Moreover, POC testing is not universally available, with rural and low-resource settings having limited availability despite their high burden of infectious diseases (22). Addressing these challenges requires targeted educational initiatives, policy support, and infrastructure investment to maximize the benefits of POC testing in antimicrobial stewardship.

## Limitations

Our study had limitations. All enrollment occurred at a single center in a high-prescribing area during the nonrespiratory season, and our control group was based on retrospective data obtained from the EHR. Future study with a concurrent control group (e.g., randomized controlled trial), multisite study, or longer cohort may be warranted. Clinicians and patients in the prospective arm of our study could opt out of participating, which could have biased our results. Self-reported data obtained in patient interviews were subject to recall bias and could have impacted our results. Our study is also at risk of the Hawthorne effect as providers could not be blinded to the intervention.

## Conclusion

Combining POC testing with an antimicrobial stewardship intervention resulted in reduced antibiotic prescribing in an urgent care setting. Implementation of POC testing

could enhance ongoing efforts to encourage appropriate prescribing practices through clinical education, guideline implementation, and stewardship programs.

## ACKNOWLEDGMENTS

We are indebted to Danielle Otmaskin and the No Resistance Consulting Group Team for their support of data collection.

This study was funded by an Investigator-Initiated Grant from bioMérieux. H.B. received funding from NIH UL1 TR001860.

L.S.M. conceived of and obtained funding for the study, participated in data analysis and interpretation, and drafting and editing the manuscript. L.S.M. takes overall responsibility for the manuscript. C.G.A. participated in data analysis and interpretation and helped draft the manuscript. T.F. participated in data analysis and helped draft and edit the manuscript. H.B. was primarily responsible for data analysis and edited the manuscript. G.H. participated in study design, was responsible for data collection, and edited the manuscript.

## AUTHOR AFFILIATIONS

[1]Department of Emergency Medicine, UC Davis School of Medicine, Sacramento, California, USA

[2]UC Davis School of Medicine, Sacramento, California, USA

[3]Department of Public Health Sciences, UC Davis School of Medicine, Davis, California, USA

[4]No Resistance Consulting Group, Birmingham, Alabama, USA

## AUTHOR ORCIDs

Cosby G. Arnold  http://orcid.org/0000-0002-8109-0460

## FUNDING

| Funder | Grant(s) | Author(s) |
| --- | --- | --- |
| National Institutes of Health | UL1 TR001860 | Heejung Bang |

## AUTHOR CONTRIBUTIONS

Cosby G. Arnold, Formal analysis | Tyra Furtado, Data curation, Formal analysis, Writing – original draft, Writing – review and editing | Heejung Bang, Formal analysis | Glenn Harnett, Data curation, Project administration | Larissa S. May, Conceptualization, Formal analysis

## DATA AVAILABILITY

The data are not publicly available.

## ADDITIONAL FILES

The following material is available online.

### Open Peer Review

**PEER REVIEW HISTORY (review-history.pdf).** An accounting of the reviewer comments and feedback.

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
