## [Reviewer comments · Microbiology Spectrum]

Microbiology Spectrum

Combining an antibiotic stewardship program with a 15-pathogen viral panel to reduce inappropriate antibiotic prescribing

Cosby Arnold, Tyra Furtado, Heejung Bang, Glenn Harnett, and Larissa May

Corresponding Author(s): Cosby Arnold, University of California Davis School of Medicine

Review Timeline:

Submission Date:	September 26, 2025
Editorial Decision:	October 14, 2025
Revision Received:	October 20, 2025
Accepted:	October 23, 2025

Editor: Eleanor Powell

Reviewer(s): The reviewers have opted to remain anonymous.

Transaction Report:

DOI: <https://doi.org/10.1128/spectrum.02195-25>

Re: Spectrum02195-25 (Combining an antibiotic stewardship program with a 15-pathogen viral panel to reduce inappropriate antibiotic prescribing)

Dear Dr. Cosby Arnold:

Thank you for the privilege of reviewing your work. After reviewing your revised JCM submission and the response to reviewer comments, I am pleased to inform you that your manuscript has been editorially accepted for publication. However, there are a few additional questions in the submission form that need to be answered before the final decision. Once these are completed, please return your submission so that I can move your paper forward to acceptance.

Revision Guidelines

Sincerely,
Eleanor Powell
Editor
Microbiology Spectrum

Re: Spectrum02195-25R1 (Combining an antibiotic stewardship program with a 15-pathogen viral panel to reduce inappropriate antibiotic prescribing)

Dear Dr. Cosby Arnold:

I'm pleased to inform you that your manuscript has been accepted, and I am forwarding it to the ASM production staff for publication. Your paper will first be checked to make sure all elements meet the technical requirements. ASM staff will contact you if anything needs to be revised before copyediting and production can begin. Otherwise, you will be notified when your proofs are ready to be viewed.

Sincerely,
Eleanor Powell
Editor
Microbiology Spectrum